# Diagnostic routes and time intervals for patients with colorectal cancer in 10 international jurisdictions; findings from a cross-sectional study from the International Cancer Benchmarking Partnership (ICBP)

David Weller, Usha Menon,[1] Alina Zalounina Falborg,[2] Henry Jensen,[2] Andriana Barisic,[3] Anne Kari Knudsen,[4] Rebecca J Bergin,[5,6] David H Brewster,[7,8] Victoria Cairnduff,[9] Anna T Gavin,[9] Eva Grunfeld,[10] Elizabeth Harland,[11] Mats Lambe,[12] Rebecca-Jane Law,[13] Yulan Lin,[4] Martin Malmberg,[14] Donna Turner,[15] Richard D Neal,[13,16] Victoria White,[5,17] Samantha Harrison,[18] Irene Reguilon,[18] ICBP Module 4 Working Group, Peter Vedsted[2]

UM and PV contributed equally.

For numbered affiliations see end of article.

**Correspondence to**
Professor David Weller;
david.weller@ed.ac.uk

## ABSTRACT

**Objective** International differences in colorectal cancer (CRC) survival and stage at diagnosis have been reported previously. They may be linked to differences in time intervals and routes to diagnosis. The International Cancer Benchmarking Partnership Module 4 (ICBP M4) reports the first international comparison of routes to diagnosis for patients with CRC and the time intervals from symptom onset until the start of treatment. Data came from patients in 10 jurisdictions across six countries (Canada, the UK, Norway, Sweden, Denmark and Australia).

**Design** Patients with CRC were identified via cancer registries. Data on symptomatic and screened patients were collected; questionnaire data from patients' primary care physicians and specialists, as well as information from treatment records or databases, supplemented patient data from the questionnaires. Routes to diagnosis and the key time intervals were described, as were between-jurisdiction differences in time intervals, using quantile regression.

**Participants** A total of 14 664 eligible patients with CRC diagnosed between 2013 and 2015 were identified, of which 2866 were included in the analyses.

**Primary and secondary outcome measures** Interval lengths in days (primary), reported patient symptoms (secondary).

**Results** The main route to diagnosis for patients was symptomatic presentation and the most commonly reported symptom was 'bleeding/blood in stool'. The median intervals between jurisdictions ranged from: 21 to 49 days (patient); 0 to 12 days (primary care); 27 to 76 days (diagnostic); and 77 to 168 days (total, from first symptom to treatment start). Including screen-detected cases did not significantly alter the overall results.

**Conclusion** ICBP M4 demonstrates important differences in time intervals between 10 jurisdictions internationally.

## Strengths and limitations of this study

► This is the first international study of this scale to use standardised survey methods to systematically examine key intervals from patients first noticing symptoms or bodily changes until the start of treatment for their colorectal cancer.

► Questionnaire data were enriched and validated with registry data (cancer registry and screening programmes) and data rules were applied consistently to ensure validity.

► As with all questionnaire based studies, there may be some response differences due to participant interpretation, cohort characteristics and sampling strategy, but we did not find obvious differences between study participants which could bias our results.

► While our analyses adjusted for age, gender and co-morbidity, we were unable to adjust for ethnicity and education due to different classification systems in participating countries.

► Understanding variations in diagnostic and treatment intervals for patients with colorectal cancer may, in jurisdictions with longer intervals, signal the need for improvements in service configuration and patient pathways.

The differences may justify efforts to reduce intervals in some jurisdictions.

## BACKGROUND

Colorectal cancer (CRC) is a leading cause of morbidity and mortality and places a major burden on health systems; worldwide

1.36 million new cases are diagnosed every year.[1] CRC is the second most common cause of death due to cancer in Europe after female breast cancer, accounting for more than 200 000 deaths per year.[2] Prognosis strongly depends on stage at diagnosis, and the disease can mostly be cured if diagnosed at an early stage. Survival has increased over the last several years in Europe.[3]

However, there remains substantial international variation in both 1-year and 5-year survival, with countries such as the UK and Denmark having significantly poorer survival than other countries such as Sweden, Canada and Australia (figure 1).[4] Some of the variation derives from differences in stage at diagnosis which, in turn, is a result of the pathway to diagnosis and treatment.[5]

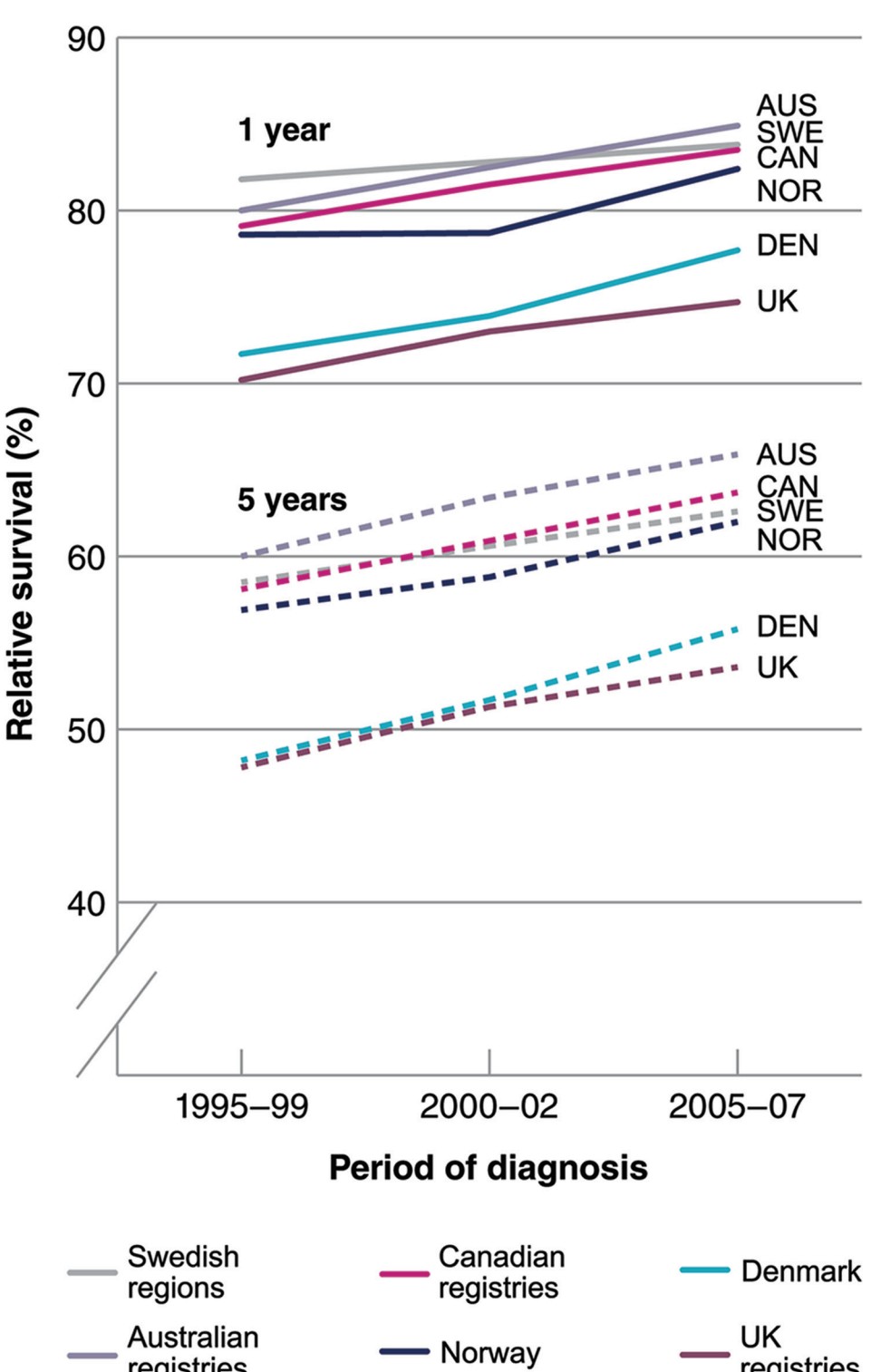

**Figure 1** Survival differences for colorectal cancer demonstrated in the International Cancer Benchmarking Partnership Module 1.[4]

Therefore, it is crucial to investigate international differences in this pathway for CRC.

The International Cancer Benchmarking Partnership (ICBP) aims to not only quantify survival differences in comparable countries but to explore factors which may impact on observed differences.[6] Module 4 (ICBP M4) focusses on the routes to diagnosis and length of diagnostic and treatment intervals as a means of understanding differences in cancer prognosis between countries. This may help shape policy and practice interventions in participating jurisdictions.[6]

Diagnosis of CRC can be difficult; the symptoms are often vague (eg, fatigue and non-specific abdominal pain), and this poses a significant diagnostic challenge for primary care, where most patients with CRC present.[7–12] There is growing evidence that prolonged diagnostic and treatment intervals are associated with poorer outcomes in CRC.[13 14] Access to investigations such as flexible sigmoidoscopy and colonoscopy is a further key issue; open access may expedite diagnosis and effect short diagnostic intervals.[12 15]

Many countries have implemented screening—typically faecal occult blood test (FOBT)-based—which can make a significant contribution to improved CRC outcomes.[16] However, currently, the large majority of CRC diagnoses are based on symptomatic presentation—for example, seeking help in primary care or attending emergency services.[7 17]

This study aims to systematically compare the diagnostic routes and time intervals from first noticing symptoms to start of treatment in patients with CRC in 10 healthcare systems with broadly similar access to high quality treatment and valid cancer registration.[6]

## METHODS

The methods for ICBP M4 have been described.[18] In brief, we recruited patients through cancer registries in 10 jurisdictions: Victoria (Australia); Manitoba and Ontario (Canada); Denmark; Norway; Sweden; England, Scotland, Northern Ireland and Wales (the UK). The target was to recruit 200 symptomatic recently diagnosed patients with CRC per jurisdiction and to measure the patient, primary care, diagnostic, treatment and total intervals (figure 2).

In defining these intervals we used principles articulated in the Aarhus Statement.[19] Data were collected from patients, their primary care physician (PCP) and their cancer treatment specialists (CTSs) as well as cancer registries. When calculating the route and time intervals we used predefined rules including a data 'hierarchy' around these information sources (online supplementary file 1). Based on a standardised protocol, teams within each jurisdiction established data collection processes with registries; survey logistics and data management were adapted to each local setting.

Data were transferred in anonymised format to the analysis team at Aarhus University—all data sources were combined into a single database.

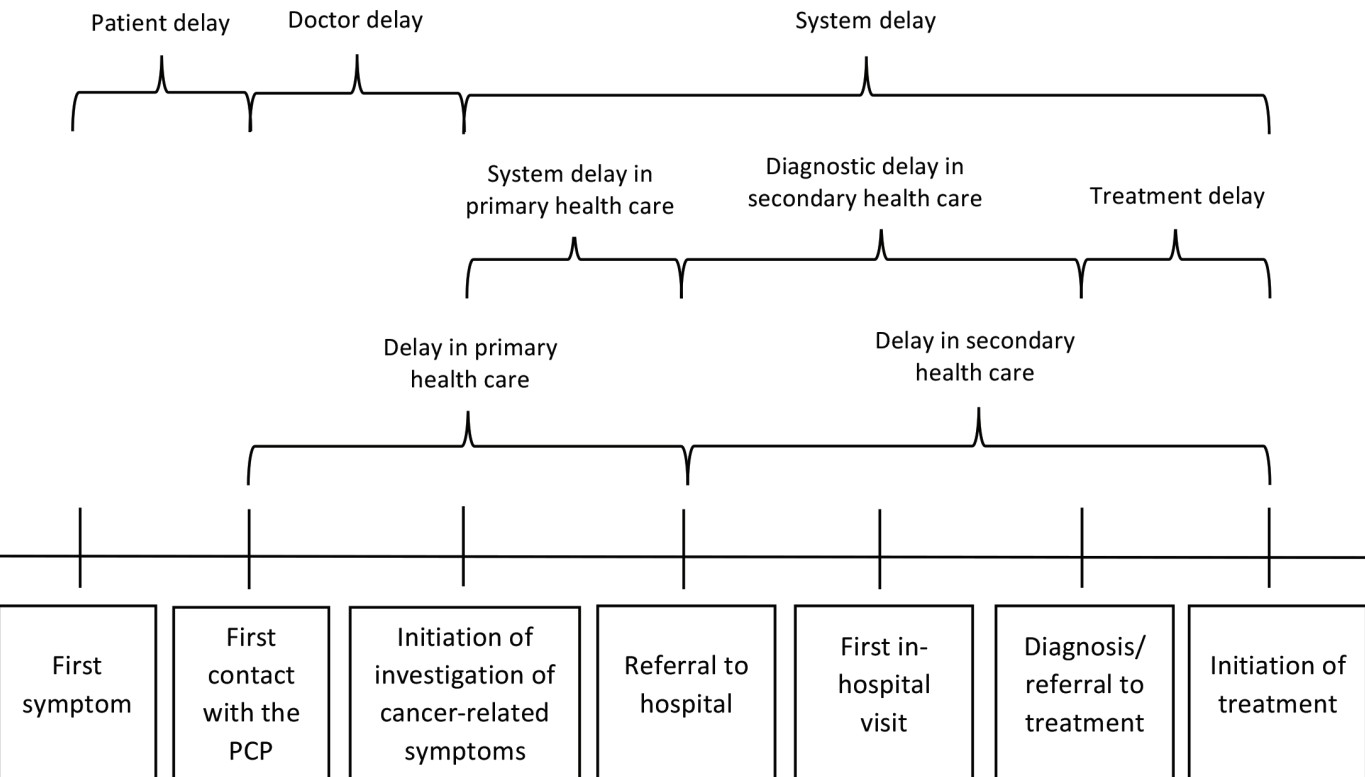

**Figure 2** Diagnosis and treatment of colorectal cancer: illustration of key time points and intervals.[19] PCP, primary care physician.

### Identification of study population

Eligible patients were consecutive patients aged 40 years or more with a first-diagnosis of CRC, International Classification of Diseases (ICD) 10 coded as C18.0-C18.9, C20.0 and C20.9.[20] Patients who had had another non-index cancer earlier were eligible, but those with synchronous different primary cancers were excluded.[18]

Each jurisdiction used a registry-based identification to enhance validity. We aimed to recruit patients 3–6 months after diagnosis; this avoided approaching patients too soon after diagnosis, while minimising recall bias from a long period post-diagnosis.

Recruitment was via cancer registries; either through (1) sending a letter to the relevant healthcare professional, requesting a preaddressed envelope be forwarded to the patient on confirmation the person was aware of the diagnosis or (2) the local research team or registry directly sending a letter to the patient.[18] Consent was required from all patients prior to surveys being sent to PCPs and CTSs and for data transfer.

### Data sources

Data from three questionnaires of eligible patients, their PCPs and CTSs (online supplementary files 2-4) were combined with information from participating cancer registries. Development, validation and implementation of these surveys is explained elsewhere.[18]

### Survey data

Questionnaires were developed collaboratively with all jurisdictions. For consented patients, based on practice lists or the patient's response, a questionnaire was sent to the PCP with whom they were listed or who had been primarily involved in the diagnostic pathway. The patients and PCPs were asked about milestones, symptoms and route to cancer diagnosis. A questionnaire was sent to the CTSs who were first involved in the treatment. Jurisdictional differences in local recruitment processes are detailed in online supplementary files 5.

### Registry data

To enhance complete and valid data on date of diagnosis, stage and screening status, data were collected through cancer registries wherever possible. Date of diagnosis was defined based on an established International Agency for Research on Cancer (IARC) hierarchy and stage was preferably given in tumour, node and metastasis and Duke's.[21 22]

### Data handling

Local teams entered data and questionnaire responses. The records were checked for obvious errors (eg, dates in future) and queries were discussed and resolved with local contacts, who checked responses against original survey. All survey data underwent cleaning centrally (Aarhus University) to ensure that the same explicit rules were applied on the full dataset. Patients where age, date of diagnosis or date of consent were unknown were excluded.

As described, the data rules allowed the combination of data from different sources in a standardised way that ensured reproducibility and transparency (online supplementary files 1). The rules, based on the Aarhus Statement,[19] employed a 'hierarchy' principle in terms of the order in which data sources (patient, PCP, CTS, registry) should take precedence where responses between sources differed, and included imputation rules based on the available data. The exact rule was guided by the measure in question—for example, patient interval was collected primarily from the patient questionnaire whereas primary care time-points were collected from the PCP questionnaire. All the measures were further validated using algorithms for outliers and out of range responses (eg, negative time intervals).

Although the protocol mandated contacting patients within a 3–6-month time window after diagnosis, some local registries needed to extend this period, primarily due to delays in recording the cancer diagnosis.

### Measures of routes to diagnosis

We defined routes to diagnosis for CRC using categories derived from the Aarhus Statement check-list—the following categories were used in the analysis[19]:
▶ Screening.
▶ Symptomatic:
  – Visit PCP.
  – Visit PCP and accident and emergency (A&E).
  – A&E.
  – Investigation for another problem.
▶ Other/unknown routes to diagnosis.

### Measures of time intervals

To ensure international comparison, the time interval definition was adapted from the Aarhus Statement and included the following time-points[19]:
▶ First onset of symptoms: the time-point when first bodily change(s) and/or symptom(s) are noticed by the patient.
▶ First presentation to healthcare: the time-point at which it would be at least possible for the clinician seeing the patient to have started investigating.
▶ First referral to secondary care: the time-point at which the PCP refers the patient (and responsibility of the patient) to secondary/specialist care.
▶ Date of diagnosis: date the definite diagnosis was made, defined by the IARC hierarchy.[21]
▶ Date of start of treatment: the date where the patient started curative or palliative treatment or a decision not to treat.

The time intervals were calculated as the number of days between these time-points (figure 2). For screen detected CRC, the patient and primary care interval were not applicable, with other intervals calculated using screening date as the first time-point. All time-points were validated manually and negative intervals were set to 0 days. Missing day was imputed based on specific rules to ensure that the direction of a possible

misclassification bias was known (online supplementary files 1).

## Establishing screening status

Patients with CRC were categorised using data rules as 'screen-detected', 'symptomatic' or 'other presentation'. In some jurisdictions it was possible to identify screen-detected cancers from registries; in others this categorisation depended on questionnaire responses. Due to differences in the understanding and registration of screening across jurisdictions, we specified symptom-based detection should include all patients who reported symptoms or A&E/primary care presentation, even if the patient had indicated 'screening' as the diagnostic route (unless their PCP or CTS specified a screening route). For UK countries the distinction between a screen-detected and non-screen-detected CRC was validated using registry data on screen-detected cases identified through public programmes.

## Covariates

Health status was measured using the self-reported general health item from the 36-Item Short Form Health Survey.[23] Comorbidity was assessed from the patient survey as presence of four diseases (stroke, diabetes, lung or heart diseases) and categorised into: 'none', 'medium' (one or two) or 'high' (three or four). Educational level was categorised as 'low' (vocational school or lower) and 'high' (university or higher). Symptoms reported were divided into two categories: 'a CRC specific symptom' or 'other symptoms'. This was based on a symptom coding done independently by two PCP-authors (DW and PV) with the aim of identifying symptoms where clinical suspicion could be raised.[24]

## Statistical analysis

Quantile regression was used to estimate differences in intervals between all jurisdictions.[25] We compared the 50th, 75th and 90th percentiles. Wales was used as the reference jurisdiction as it had the lowest CRC survival according to the ICBP Module 1 cancer survival benchmark.[4] Quantile regression allows a comparison on the interval scale with optimal information on differences. Counting days, we used the 'qcount' procedure proposed by Miranda.[26] Parameters were calculated with 1000 jittered samples. The differences (in days) in intervals between jurisdictions (using Wales as a reference) were calculated as marginal effects after quantile regression by setting the continuous covariate age to its mean value and the categorical covariates (gender and comorbidity) to their modes. Significance level was set to 0.05 or less, and 95% CI were calculated when appropriate. Statistical analyses were carried out using STATA V.14 software.

## Sensitivity and validity analyses

All analyses were undertaken using the 6 and 9 months cut-off criteria for allowable interval from diagnosis to questionnaire completion. To estimate the effect of using patient reported intervals only, a sensitivity analysis based solely on patient data was performed. The effect of excluding patients for whom at least one time interval hadn't been reported was also investigated.

Kappa coefficient and overall agreement percentage assessed the agreement on routes to diagnosis (screening and symptomatic presentation) between the different data sources. Kappa coefficients were interpreted using Landis' and Koch's criteria[27]: 0.00–0.20=slight, 0.21–0.40=fair, 0.41–0.60=moderate, 0.61–0.80=substantial, above 0.80=almost perfect.

Agreement between the different data sources was also assessed by Lin's concordance correlation coefficient (CCC).[28] The ICBP M4 definition of screening-status was validated against registry data on screening where available, and assessed by kappa and overall agreement percentage.

## Patient involvement

The research questions for this survey drew on an extensive literature elating diagnosis and treatment delays leading to negative patient experiences. While patient experience was not a primary outcome measure for this study, patients were given the opportunity to comment on their experience through questionnaire free-text response options (under separate analysis). Patients were involved in the piloting of study instruments to ascertain if recruitment and questionnaire content and dissemination strategies were appropriate, described elsewhere.[18] Each jurisdiction has committed to communicating the findings and local implications of this study to organisations representing their study participants.

## RESULTS

### Patient characteristics and participation

Of 14 664 eligible patients, 3881 returned completed questionnaires (a 31% response rate, ranging from 19% in Norway to 69% in Denmark). Of these, 2866 (95%) were included in the analyses after application of inclusion and exclusion criteria. The study flow with identification, exclusion and responses for each jurisdiction is seen in table 1.

The characteristics of the included patients are detailed in table 2. The patient questionnaire was completed at a median of 5 months following diagnosis. For tumour stage the proportion of missing stage was high in Norway due to lack of registry data.

### Routes to diagnosis

Routes to diagnosis were broadly similar, except for screening; of all patients, 16.1% had a screen detected CRC, ranging from 6.3% in Norway to 31.4% in Manitoba (table 3). In Denmark and Sweden CRC-screening had not been implemented at the time of study and screening status in Norway was determined by information from local screening trials. The proportion of screen-detected cancers in Northern Ireland is lower as most were excluded in the recruitment process, with the local team

**Table 1** Patient flow from identification to analyses for all 10 jurisdictions

| Jurisdiction | Wales | England | Scotland | Northern Ireland | Denmark | Manitoba | Norway | Sweden | Ontario | Victoria | Total |
|---|---|---|---|---|---|---|---|---|---|---|---|
| Start date | 04/10/2013 | 01/11/2013 | 01/12/2013 | 06/08/2013 | 28/10/2013 | 01/05/2013 | 01/09/2014 | 01/02/2014 | 30/04/2014 | 01/07/2013 | |
| | n (%) | n (%) | n (%) | n (%) | n (%) | n (%) | n (%) | n (%) | n (%) | n (%) | n (%) |
| **Eligible patients*†** | 1274 (100) | 1314 (100) | 1852 (92.4) | 568 (45.0) | 490 (79.9) | 1288 (84.6) | 1860 (95.5) | 537 (85.8) | 5585 (71.8)‡ | 1170 (58.7) | 14664 (76.9) |
| **Packs sent to PCP§** | 1274 (100) | 1198 (91.2) | 1070 (57.8) | | | | | | | | 3542 (79.8) |
| Pack not forwarded by PCP | 211 (16.6) | 87 (7.3) | 103 (9.6) | | | | | | | | 401 (11.3) |
| Unsure if pack forwarded by PCP | 333 (26.1) | 362 (30.2) | 209 (19.5) | | | | | | | | 904 (25.5) |
| Pack forwarded by PCP | 730 (57.3) | 749 (62.5) | 758 (70.8) | | | | | | | | 2237 (63.2) |
| **Patients contacted by PCP§¶** | 1063 (83.4) | 1111 (92.7) | 967 (90.4) | | | | | | | | 3141 (88.7) |
| **Patients approached directly§** | | | | 555 (97.7) | 490 (100) | 761 (59.1) | 1860 (100) | 537 (100) | 5099 (91.3) | 1049 (89.7) | 10351 (70.6) |
| Patient died | | | | | | 49 (3.2) | | | 139 (1.8) | | 188 (1.8) |
| Other | | | | 13 (1.0) | | 26 (1.7) | | | 368 (4.7) | | 407 (3.9) |
| No address | | | | | | 11 (0.7) | | | 309 (4.0) | | 320 (3.1) |
| **Patient responses (% of eligible patients)§** | 314 (24.6) | 285 (21.7) | 337 (18.2) | 283 (49.8) | 340 (69.4) | 274 (21.3) | 358 (19.2) | 319 (59.4) | 899 (16.1) | 472 (40.3) | 3881 (26.5) |
| **Patient responses (% of contacted)**** | 314 (29.5) | 285 (25.7) | 337 (34.9) | 283 (52.2) | 340 (69.4) | 274 (40.6) | 358 (19.2) | 319 (59.4) | 899 (21.0) | 472 (45.0) | 3881 (30.9) |
| Did not fulfil eligibility criteria | 7 (2.2) | 1 (0.4) | | 1 (0.4) | | 1 (0.4) | | | | | 10 (0.3) |
| Received after submission | | | | 20 (7.1) | | | | | | 55 (11.7) | 75 (2.1) |
| Other | 7 (2.2) | | 57 (16.9) | 6 (2.1) | 16 (4.7) | 11 (4.0) | | | 45 (5.0) | 127 (26.9) | 269 (7.6) |
| **Patient surveys forwarded for analyses††** | 300 (95.5) | 284 (99.6) | 280 (83.1) | 256 (90.5) | 324 (95.3) | 262 (95.6) | 358 (100) | 319 (100) | 854 (95.0) | 290 (61.4) | 3527 (90.9) |
| *Excluded for analyses—total* | *17 (5.7)* | *10 (3.5)* | *7 (2.5)* | *3 (1.2)* | *7 (2.2)* | *4 (1.5)* | *72 (20.1)* | *8 (2.5)* | *532 (62.3)* | *1 (0.3)* | *661 (18.7)* |
| *Not sampled* | *0 (0)* | *0 (0)* | *0 (0)* | *0 (0)* | *0 (0)* | *0 (0)* | *0 (0)* | *0 (0)* | *494 (57.8)* | *0 (0)* | *494 (14)* |
| *Previous cancer* | *0 (0)* | *3 (1.1)* | *0 (0)* | *0 (0)* | *0 (0)* | *0 (0)* | *0 (0)* | *0 (0)* | *1 (0.1)* | *0 (0)* | *4 (0.1)* |
| *Unknown date of consent* | *0 (0)* | *0 (0)* | *0 (0)* | *0 (0)* | *0 (0)* | *0 (0)* | *3 (0.8)* | *6 (1.9)* | *0 (0)* | *0 (0)* | *9 (0.3)* |
| *Unknown date of diagnosis* | *0 (0)* | *0 (0)* | *0 (0)* | *0 (0)* | *0 (0)* | *1 (0.4)* | *6 (1.7)* | *0 (0)* | *0 (0)* | *0 (0)* | *7 (0.2)* |
| *Consent too late/too early* | *17 (5.7)* | *7 (2.5)* | *7 (2.5)* | *3 (1.2)* | *0 (0)* | *3 (1.1)* | *60 (16.8)* | *1 (0.3)* | *37 (4.3)* | *1 (0.3)* | *136 (3.9)* |
| *Other* | *0 (0)* | *0 (0)* | *0 (0)* | *0 (0)* | *7 (2.2)* | *0 (0)* | *3 (0.8)* | *1 (0.3)* | *0 (0)* | *0 (0)* | *11 (0.3)* |
| Patient surveys analysed (% of forwarded surveys) | 283 (94.3) | 274 (96.5) | 273 (97.5) | 253 (98.8) | 317 (97.8) | 258 (98.5) | 286 (79.9) | 311 (97.5) | 322 (89.4) | 289 (99.7) | 2866 (94.5)‡‡ |
| PCP surveys analysed (% of analysed patients) | 234 (82.7) | 225 (82.1) | 224 (82.1) | 213 (84.2) | 241 (76.0) | 148 (57.4) | 169 (59.1) | n/a | 121 (37.7) | 199 (68.9) | 1774 (69.5)§§ |
| Specialist surveys analysed (% of analysed patients) | 164 (58.0) | 156 (56.9) | 179 (65.6) | n/a¶¶ | 187¶¶ (59) | n/a*** | 64 (22.4) | n/a*** | 89 (27.7) | 99 (34.3) | 938 (45.9)††† |

Continued

## Table 1 Continued

| Jurisdiction | Wales | England | Scotland | Northern Ireland | Denmark | Manitoba | Norway | Sweden | Ontario | Victoria | Total |
|---|---|---|---|---|---|---|---|---|---|---|---|
| Start date | 04/10/2013 | 01/11/2013 | 01/12/2013 | 06/08/2013 | 28/10/2013 | 01/05/2013 | 01/09/2014 | 01/02/2014 | 30/04/2014 | 01/07/2013 | |
| | n (%) | n (%) | n (%) | n (%) | n (%) | n (%) | n (%) | n (%) | n (%) | n (%) | n (%) |

*Eligible according to protocol: that is, woman, 40 years or more, alive, consented to participate within 9 months of diagnosis, diagnosed with breast cancer (ICD-10: C50.0-C50.9), behaviour code ICD-O-3=3 and without prior history of cancer of the breast or synchronous primary breast cancer.

†In some jurisdictions, some 'eligible' patients had preopted out of being contacted and in a small number PCP information was not available.

‡initially, 1458 additional cases were eligible but excluded for this study as urban sample size was reached.

§Percentages of eligible patients.

¶Maximum of potentially contacted patients, that is, sum of packs forwarded by PCP and packs unsure if forwarded by PCP.

**Percentages of patients contacted by PCP (see note ¶) for Wales, England and Scotland or percentages of patients contacted directly by a registry excl. non-accessible patients (all other jurisdictions).

††Percentages of patient responses.

‡‡Denominator=total number of forwarded cases excl. patients not included in analytic sample in Ontario.

§§Denominator=total number of analysed cases excl. patients from Sweden.

¶¶Data obtained from registries instead: Northern Ireland from the Northern Ireland Cancer Registry, supported by extracts from clinical datasets; Denmark from the Danish Colorectal Cancer Group.

***Data not collected in this jurisdiction.

†††Denominator=total number of analysed cases excl. patients from Sweden, Manitoba and Northern Ireland.

ICD, International Classification of Diseases; PCP, primary care physician.

---

actively including symptomatic patients in order to reach the target of 200 symptomatic patients. Overall (excluding Northern Ireland), most (82.2%) respondents presented with a symptomatic CRC. A high level of agreement was found between ICBP and registry data for screening status (table 3). PCP data indicated that the percentage of the symptomatic patients urgently referred with a suspicion of cancer was less than 50% in Ontario, Denmark, Manitoba, Norway (37.8%, 39.6%, 46.8%, 47.8%, correspondingly) and larger than 50% in Scotland, Northern Ireland, Victoria, England, Wales (51.5%, 57.1%, 57.6%, 67.6%, 69.3%, correspondingly).

### Symptom prompting concern

The proportion (%) of patients and PCPs reporting symptoms are shown in table 4 (for Northern Ireland and Sweden, only patient data were collected). The most common symptom reported by PCP respondents was rectal bleeding (40% of respondents), followed by change in bowel habit. While every third patient indicated fatigue as a key symptom, it was rarely reported by PCPs.

### Time intervals

The median patient interval varied from 21 days (Denmark) to 49 days (Wales) (table 5). Table 6 shows the adjusted patient median interval was 25 days shorter in Denmark than in Wales; none were longer compared with Wales (online supplementary files 6).

The median primary care interval was 12 days in Norway (table 5), statistically significantly longer than Wales (table 6). For the 10% of patients waiting longest for referral, the longest intervals were observed in Manitoba, Victoria and Scotland (210, 128 and 93 days, respectively) (table 5). This interval at the 90th percentile was either 4 months (Manitoba) or 1 month (Victoria and Scotland) longer than in Wales (table 6).

The median diagnostic interval for symptomatic patients ranged from 27 to 28 days in Denmark and Victoria to 76 days in Manitoba. At the 90th percentile it ranged from 4 months in Denmark to 10 months in Ontario (table 5). All jurisdictions had shorter median diagnostic intervals compared with Wales, except Northern Ireland and Manitoba, where the intervals were 5 and 14 days longer, respectively (table 6).

The shortest median treatment intervals for all patients (about 2 weeks) were observed in Denmark, Victoria and Norway. In other jurisdictions this interval was 1 month or more (table 5). All jurisdictions had shorter treatment intervals compared with Wales, except Scotland and Manitoba (table 6).

The median total interval (from first symptom to treatment start) for all patients with CRC was between 74 days (Victoria) and 136 days (Manitoba) (table 5). In Scotland, Denmark, Norway and Victoria, this interval was statistically significantly shorter than in Wales (table 6).

**Table 2** The characteristics of the 2866 patients aged 40 or over with a first diagnosis of CRC included in the analyses (% if nothing else stated)

| | Wales (N=283) | England (N=274) | Scotland (N=273) | Northern Ireland (N=253) | Denmark (N=317) | Manitoba (N=258) | Norway (N=286) | Sweden (N=311) | Ontario (N=322) | Victoria (N=289) | Total (N=2866) |
|---|---|---|---|---|---|---|---|---|---|---|---|
| Median (range) interval for diagnosis to questionnaire completion in months | 5 (3–9) | 5 (1–9) | 5 (3–9) | 4 (2–6) | 5 (0.5–8) | 6 (3–9) | 7 (0.03–9) | 4 (0.5–7) | 6 (1–9) | 6 (3–9) | 5 (0.03–9) |
| Age years | | | | | | | | | | | |
| Median (IQI) | 71 (65–79) | 71 (64–78) | 70 (61–77) | 67 (60–74) | 71 (65–77) | 69 (59–77) | 70 (62–77) | 72 (65–79) | 69 (61–77) | 66 (58–76) | 70 (62–77) |
| Age years (symptomatic patients) | (N=208) | (N=212) | (N=192) | (N=214) | (N=311) | (N=176) | (N=264) | (N=307) | (N=257) | (N=220) | (N=2361) |
| Median (IQI) | 72 (64–81) | 72 (64–80) | 72 (61–80) | 67 (59–76) | 71 (65–77) | 73 (60–81) | 70 (62–77) | 72 (65–79) | 69 (61–77) | 69 (57–77) | 71 (62–78) |
| Age years (screen-detected patients) | (N=73) | (N=58) | (N=78) | (N=35) | (N=0) | (N=81) | (N=18) | (N=0) | (N=55) | (N=62) | (N=460) |
| Median (IQI) | 67 (65–73) | 68 (65–74) | 66 (62–72) | 68 (63–70) | n/a | 65 (57–70) | 69 (61–74) | n/a | 68 (63–72) | 65 (60–69) | 67 (63–72) |
| Gender | | | | | | | | | | | |
| Male | 59 | 56.2 | 58.6 | 58.9 | 59 | 56.6 | 51.4 | 51.1 | 56.5 | 56.1 | 56.3 |
| Health state | | | | | | | | | | | |
| Good | 80.6 | 84.7 | 84.6 | 75.5 | 82.3 | 88 | 74.1 | 78.8 | 85.1 | 82.4 | 81.6 |
| Fair | 13.8 | 11.7 | 10.6 | 17.4 | 13.6 | 9.3 | 22 | 16.1 | 11.5 | 12.1 | 13.8 |
| Poor | 5 | 3.3 | 3.3 | 6.7 | 1.6 | 1.9 | 2.8 | 3.2 | 2.5 | 4.2 | 3.4 |
| Missing | 0.7 | 0.4 | 1.5 | 0.4 | 2.5 | 0.8 | 1.1 | 1.9 | 0.9 | 1.4 | 1.2 |
| Comorbidity* | | | | | | | | | | | |
| No | 52.7 | 58.8 | 57.1 | 55.3 | 50.2 | 63.2 | 61.9 | 58.5 | 53.7 | 61.9 | 57.2 |
| Medium | 44.2 | 38.7 | 41.8 | 43.5 | 46.7 | 34.9 | 35 | 35.1 | 31.4 | 35.6 | 38.6 |
| High | 2.5 | 2.6 | 0.7 | 1.2 | 1.9 | 1.6 | 3.2 | 4.5 | 1.9 | 1.7 | 2.2 |
| Missing | 0.7 | 0 | 0.4 | 0 | 1.3 | 0.4 | 0 | 1.9 | 13 | 0.7 | 2 |
| Education | | | | | | | | | | | |
| Low | 76.7 | 80.7 | 74.7 | 77.5 | 76 | 80.2 | 75.2 | 78.8 | 73.3 | 77.5 | 77 |
| High | 15.6 | 14.2 | 19.8 | 13 | 12 | 17.8 | 18.9 | 20.3 | 23.9 | 21.5 | 17.8 |
| Missing | 7.8 | 5.1 | 5.5 | 9.5 | 12 | 1.9 | 5.9 | 1 | 2.8 | 1 | 5.2 |
| Ethnicity | | | | | | | | | | | |
| White | 99.9 | 98.5 | 98.5 | 99.6 | 95.9 | 93.4 | 99.7 | 99.4 | 92.6 | 94.5 | 97.1 |
| Asian | 0.4 | 0.4 | 0.7 | 0 | 0.3 | 1.9 | 0 | 0.3 | 5.9 | 2.4 | 1.3 |
| Black | 0 | 0 | 0.4 | 0 | 0 | 0.4 | 0 | 0 | 0.6 | 0 | 0.1 |
| Other | 0 | 0 | 0 | 0 | 0 | 3.5 | 0 | 0 | 0.3 | 0 | 0.4 |
| Missing | 0 | 1.1 | 0.4 | 0.4 | 3.8 | 0.8 | 0.3 | 0.3 | 0.6 | 3.1 | 1.1 |
| Smoking | | | | | | | | | | | |
| Currently | 4.2 | 2.6 | 8.1 | 9.1 | 11.4 | 8.9 | 7 | 4.8 | 4.4 | 4.8 | 6.5 |
| In the past | 55.5 | 54.7 | 51.3 | 49 | 55.2 | 50.8 | 56.3 | 52.7 | 59.3 | 51.6 | 53.8 |
| Never | 39.9 | 41.2 | 40.7 | 39.5 | 31.6 | 39.9 | 36.4 | 42.4 | 35.4 | 42.6 | 38.8 |

Continued

**Table 2** Continued

| | Wales (N=283) | England (N=274) | Scotland (N=273) | Northern Ireland (N=253) | Denmark (N=317) | Manitoba (N=258) | Norway (N=286) | Sweden (N=311) | Ontario (N=322) | Victoria (N=289) | Total (N=2866) |
|---|---|---|---|---|---|---|---|---|---|---|---|
| Missing | 0.4 | 1.5 | 0 | 2.4 | 1.9 | 0.4 | 0.3 | 0 | 0.9 | 1 | 0.9 |
| **Tumour stage—TNM and Duke's** | | | | | | | | | | | |
| 0 | 0.4 | 0.4 | 0 | 0 | 0 | 0 | 0 | 0 | 0.6 | 0 | 0.1 |
| I | 19.4 | 20.8 | 18.7 | 18.2 | 20.5 | 17.1 | 3.9 | 16.7 | 27 | 26 | 19 |
| II | 25.4 | 29.6 | 32.6 | 29.6 | 36 | 28.3 | 4.9 | 26.1 | 32 | 35.6 | 28.1 |
| III | 40.3 | 30.7 | 35.9 | 37.6 | 24.6 | 39.5 | 7 | 31.5 | 28 | 29.1 | 30.1 |
| IV | 9.5 | 16.1 | 12.1 | 13.4 | 13.9 | 11.6 | 4.6 | 17.4 | 6.5 | 8.3 | 11.3 |
| Missing | 5 | 2.6 | 0.7 | 1.2 | 5 | 3.5 | 79.7 | 8.4 | 5.9 | 1 | 11.4 |
| **Tumour stage—TNM and Duke's (symptomatic patients)** | (N=208) | (N=212) | (N=192) | (N=214) | (N=311) | (N=176) | (N=264) | (N=307) | (N=256) | (N=220) | (N=2360) |
| 0 | 0.5 | 0.5 | 0 | 0 | 0 | 0 | 0 | 0 | 0.4 | 0 | 0.1 |
| I | 16.4 | 17 | 13.5 | 14 | 20.9 | 15.3 | 3.8 | 16.6 | 22.6 | 20.9 | 16.2 |
| II | 26.4 | 30.7 | 34.9 | 31.8 | 35.1 | 29.6 | 4.2 | 26.4 | 33.5 | 36.8 | 28.6 |
| III | 39.9 | 31.6 | 36.5 | 38.3 | 25.1 | 36.9 | 7.2 | 31.6 | 29.6 | 30.9 | 29.9 |
| IV | 11.5 | 17 | 14.1 | 14.5 | 13.8 | 13.1 | 4.9 | 17.3 | 8.2 | 10 | 12.4 |
| Missing | 5.3 | 3.3 | 1 | 1.4 | 5.1 | 5.1 | 79.9 | 8.1 | 5.8 | 1.4 | 12.8 |
| **Tumour stage—TNM and Duke's (screen-detected patients)** | (N=73) | (N=58) | (N=78) | (N=35) | (N=0) | (N=81) | (N=18) | (N=0) | (N=55) | (N=62) | (N=460) |
| 0 | 0 | 0 | 0 | 0 | n/a | 0 | 0 | n/a | 1.8 | 0 | 0.2 |
| I | 28.8 | 36.2 | 30.8 | 40 | n/a | 21 | 5.6 | n/a | 47.3 | 47 | 33.3 |
| II | 23.3 | 24.1 | 28.2 | 17.1 | n/a | 24.7 | 16.7 | n/a | 21.8 | 29 | 24.4 |
| III | 41.1 | 27.6 | 33.3 | 34.3 | n/a | 45.7 | 5.6 | n/a | 21.8 | 21 | 32 |
| IV | 2.7 | 12.1 | 7.7 | 8.6 | n/a | 8.6 | 0 | n/a | 0 | 3.2 | 5.9 |
| Missing | 4.1 | 0 | 0 | 0 | n/a | 0 | 72.2 | n/a | 7.3 | 0 | 4.4 |

*Comorbidity coded as no=none reported, medium=1–2 reported and high=3+ reported.
CRC, colorectal cancer; IQI, inter quartile interval; n/a, not applicable; TNM, tumour, node and metastasis.

**Table 3** The overall route (symptomatic or screened) for CRC in each jurisdiction (%) and place of initial presentation for symptomatic patients

| | Wales | England | Scotland | Northern Ireland* | Denmark | Manitoba | Norway† | Sweden | Ontario | Victoria | Total |
|---|---|---|---|---|---|---|---|---|---|---|---|
| | (N=283) | (N=274) | (N=273) | (N=253) | (N=317) | (N=258) | (N=286) | (N=311) | (N=322) | (N=289) | (N=2866) |
| **Symptomatic** | **73.5** | **77.3** | **70.3** | **84.6** | **98.1** | **68.2** | **92.3** | **99** | **79.8** | **76.1** | **82.4** |
| *Visit PCP, visit PCP and A&E‡* | 77.4 | 82.6 | 92.3 | 83.2 | 82.3 | 67.1 | 78.4 | 51.8 | 62.3 | 73.2 | 73.2 |
| *A&E‡* | 7.2 | 3.8 | 0.5 | 6.5 | 3.2 | 9.1 | 4.6 | 11.7 | 8.2 | 8.6 | 6.7 |
| *Investigation for another problem‡* | 6.7 | 4.7 | 4.8 | 3.3 | 9.3 | 9.7 | 9.9 | 32.9 | 11.3 | 7.3 | 11.1 |
| *Other‡* | 8.7 | 9 | 2.4 | 7 | 5.1 | 14.2 | 7.2 | 3.6 | 18.3 | 10.9 | 8.9 |
| **Screening** | **25.8** | **21.2** | **28.6** | **13.8** | **0** | **31.4** | **6.3** | **0** | **17.1** | **21.5** | **16.1** |
| **Other** | **0.7** | **1.5** | **1.1** | **1.6** | **1.9** | **1** | **1.4** | **1** | **3.1** | **2.4** | **1.5** |

*Agreement between the ICBP M4 presentation-rule (without using registry data) and registry information to define if a CRC case was screen detected*

| Jurisdiction | Number of cases | Agreement on screening-status between ICBP M4 and registry % | Kappa | (95%CI) |
|---|---|---|---|---|
| Wales | 277 | 92 | 0.78 | (0.68 to 0.86) |
| England | 259 | 95 | 0.84 | (0.74 to 0.92) |
| Scotland | 270 | 93 | 0.83 | (0.75 to 0.90) |
| Northern Ireland | 251 | 94 | 0.75 | (0.61 to 0.87) |

*In Northern Ireland, the proportion of screen detected CRC cases was lower as these patients were primarily excluded from the eligible group.
†The reporting of screening status in Norway was based on local screening trials.
‡Percentage of symptomatic route.
A&E, accident and emergency; CRC, colorectal cancer; ICBP M4, International Cancer Benchmarking Partnership Module 4; PCP, primary care physician.
Bold values are statistically significant.

**Table 4** The symptoms experienced by patients before presentation and the presenting symptoms seen by the PCP for the 2361 patients aged 40 or over with a first diagnosis of CRC who had a symptom-based diagnosis

| | Wales | England | Scotland | Northern Ireland | Denmark | Manitoba | Norway | Sweden | Ontario | Victoria | Total |
|---|---|---|---|---|---|---|---|---|---|---|---|
| **First symptom (reported by patient)** | (N=208) | (N=212) | (N=192) | (N=214) | (N=311) | (N=176) | (N=264) | (N=307) | (N=257) | (N=220) | (N=2361) |
| Bleeding/blood in stool | 43.8 | 41 | 37.5 | 43 | 34.7 | 42 | 51.9 | 47.2 | 50.6 | 37.3 | 43.1 |
| Bowel habit change | 42.3 | 33.5 | 33.9 | 42.1 | 31.8 | 27.3 | 28.4 | 35.8 | 48.3 | 31.4 | 35.5 |
| Fatigue | 31.3 | 33.5 | 34.4 | 37.9 | 19.3 | 31.3 | 24.6 | 30 | 31.5 | 29.6 | 29.7 |
| Abdominal pain | 24 | 25 | 28.7 | 27.6 | 19.3 | 26.7 | 22 | 19.2 | 22.2 | 28.2 | 23.7 |
| Weight loss | 18.8 | 19.3 | 22.4 | 17.3 | 14.2 | 14.7 | 14 | 18.2 | 15.6 | 16.4 | 16.9 |
| Other | 40.4 | 35.9 | 37 | 42.5 | 27.3 | 34.7 | 31.1 | 40.1 | 18.7 | 30.5 | 33.4 |
| No symptoms | 6.7 | 3.8 | 9.9 | 4.7 | 9.7 | 14.8 | 7.6 | 9.5 | 5.8 | 8.2 | 8 |
| Missing | 1.4 | 1.9 | 2.6 | 1.9 | 10.3 | 3.4 | 3 | 1 | 1.2 | 0.9 | 3 |
| **Number of symptoms per patient** | | | | | | | | | | | |
| Median (IQI) | 2(1,3) | 2(1,3) | 2(1,3) | 2(1,3) | 1(1,2) | 2(1,3) | 2(1,2) | 2(1,3) | 2(1,3) | 2(1,3) | 2(1,3) |
| **Presenting symptom (reported by PCP)** | (N=140) | (N=151) | (N=132) | (N=0) | (N=212) | (N=77) | (N=134) | (N=0) | (N=82) | (N=118) | (N=1046) |
| Bleeding/blood in stool | 37.1 | 33.8 | 33.3 | n/a | 29.1 | 26 | 29.1 | n/a | 32.9 | 33.9 | 31.9 |
| Bowel habit change | 33.6 | 27.8 | 21.2 | n/a | 26.9 | 16.9 | 17.9 | n/a | 19.5 | 21.2 | 24.1 |
| Fatigue | 4.3 | 1.3 | 3.8 | n/a | 7.6 | 5.2 | 4.5 | n/a | 9.8 | 8.5 | 5 |
| Abdominal pain | 20.7 | 15.2 | 17.4 | n/a | 18.9 | 16.9 | 12.7 | n/a | 15.9 | 26.3 | 18.1 |
| Weight loss | 8.6 | 6.6 | 8.3 | n/a | 7.6 | 1.3 | 3 | n/a | 3.7 | 6.8 | 6.2 |
| Other | 36.4 | 30.5 | 36.4 | n/a | 34.9 | 27.3 | 33.6 | n/a | 18.3 | 39.8 | 33.2 |
| No symptoms | 3.6 | 4.6 | 6.8 | n/a | 0.5 | 15.6 | 0 | n/a | 3.7 | 1.7 | 3.7 |
| Missing | 8.6 | 8.6 | 9.1 | n/a | 16.5 | 26 | 8.2 | n/a | 17.1 | 11 | 12.4 |
| **Cancer-specificity of symptom presented** | | | | | | | | | | | |
| Cancer-specific symptom | 67.9 | 66.2 | 56.1 | n/a | 57.6 | 46.8 | 53 | n/a | 48.8 | 63.6 | 58.6 |
| Non-specific symptom | 20 | 20.5 | 28 | n/a | 25.5 | 11.7 | 38.8 | n/a | 30.5 | 23.7 | 25.2 |
| No symptoms /missing | 12.1 | 13.3 | 15.9 | n/a | 17 | 41.6 | 8.2 | n/a | 20.7 | 12.7 | 16.2 |

All figures are in % unless otherwise specified.
CRC, colorectal cancer; IQI, interquartile interval; n/a, not applicable; PCP, primary care physician.

**Table 5** The time intervals (days) for each of the 10 jurisdictions depicted as median, 75th and 90th percentiles

| | | Wales | England | Scotland | Northern Ireland | Denmark | Manitoba | Norway | Sweden | Ontario | Victoria |
|---|---|---|---|---|---|---|---|---|---|---|---|
| Patient interval (Symptomatic patients) | Number | (n=195) | (n=199) | (n=175) | (n=199) | (n=292) | (n=134) | (n=240) | (n=266) | (n=230) | (n=199) |
| | Median | 49 | 34 | 30 | 35 | 21 | 34 | 36 | 31 | 31 | 22 |
| | 75th percentile | 92 | 118 | 73 | 88 | 62 | 92 | 92 | 92 | 96 | 63 |
| | 90th percentile | 249 | 346 | 181 | 312 | 180 | 215 | 218 | 201 | 304 | 234 |
| Primary care interval (Symptomatic patients) | Number | (n=157) | (n=152) | (n=127) | (n=160) | (n=207) | (n=72) | (n=124) | (n=0) | (n=77) | (n=117) |
| | Median | 3 | 2 | 4 | 0 | 1 | 4 | 12 | n/a | 1 | 9 |
| | 75th percentile | 20 | 21 | 28 | 14 | 10 | 30 | 39 | n/a | 23 | 32 |
| | 90th percentile | 78 | 54 | 93 | 54 | 51 | 210 | 82 | n/a | 70 | 128 |
| Diagnostic interval (Symptomatic patients) | Number | (n=194) | (n=196) | (n=174) | (n=190) | (n=290) | (n=133) | (n=229) | (n=249) | (n=218) | (n=197) |
| | Median | 60 | 48 | 38 | 64 | 27 | 76 | 37 | 36 | 54 | 28 |
| | 75th percentile | 155 | 86 | 91 | 111 | 66 | 162 | 85 | 82 | 146 | 66 |
| | 90th percentile | 284 | 201 | 164 | 238 | 129 | 365 | 222 | 196 | 312 | 200 |
| Diagnostic interval (Screen-detected patients) | Number | (n=69) | (n=56) | (n=76) | (n=35) | (n=0) | (n=25) | (n=14) | (n=0) | (n=50) | (n=38) |
| | Median | 35 | 25 | 36 | 0 | n/a | 66 | 22 | n/a | 3 | 40 |
| | 75th percentile | 65 | 46 | 49 | 0 | n/a | 111 | 48 | n/a | 43 | 64 |
| | 90th percentile | 99 | 70 | 76 | 0 | n/a | 206 | 84 | n/a | 120 | 122 |
| Diagnostic interval (All patients) | Number | (n=263) | (n=252) | (n=250) | (n=225) | (n=290) | (n=158) | (n=243) | (n=249) | (n=268) | (n=235) |
| | Median | 52 | 43 | 37 | 47 | 27 | 72 | 36 | 36 | 44 | 28 |
| | 75th percentile | 120 | 76 | 72 | 101 | 66 | 139 | 85 | 82 | 128 | 64 |
| | 90th percentile | 242 | 176 | 151 | 207 | 129 | 320 | 212 | 196 | 278 | 178 |
| Treatment interval (Symptomatic patients) | Number | (n=197) | (n=206) | (n=185) | (n=208) | (n=306) | (n=161) | (n=258) | (n=281) | (n=248) | (n=209) |
| | Median | 39 | 31 | 33 | 25 | 14 | 34 | 18 | 35 | 33 | 14 |
| | 75th percentile | 59 | 47 | 56 | 40 | 19 | 59 | 29 | 52 | 54 | 29 |
| | 90th percentile | 83 | 60 | 79 | 58 | 28 | 97 | 45 | 65 | 79 | 47 |
| Treatment interval (Screen-detected patients) | Number | (n=72) | (n=58) | (n=78) | (n=34) | (n=0) | (n=79) | (n=17) | (n=0) | (n=52) | (n=60) |
| | Median | 44 | 39 | 49 | 38 | n/a | 38 | 19 | n/a | 40 | 17 |
| | 75th percentile | 68 | 46 | 71 | 52 | n/a | 61 | 27 | n/a | 54 | 35 |
| | 90th percentile | 80 | 62 | 91 | 61 | n/a | 83 | 43 | n/a | 88 | 44 |
| Treatment interval (All patients) | Number | (n=271) | (n=268) | (n=266) | (n=246) | (n=312) | (n=240) | (n=279) | (n=284) | (n=310) | (n=276) |
| | Median | 41 | 34 | 37 | 27 | 14 | 35 | 18 | 36 | 34 | 15 |
| | 75th percentile | 63 | 47 | 63 | 42 | 19 | 60 | 28 | 53 | 54 | 29 |
| | 90th percentile | 80 | 61 | 87 | 59 | 27 | 88 | 43 | 65 | 82 | 44 |

Continued

**Table 5** Continued

| | | Wales | England | Scotland | Northern Ireland | Denmark | Manitoba | Norway | Sweden | Ontario | Victoria |
|---|---|---|---|---|---|---|---|---|---|---|---|
| Total interval (Symptomatic patients) | Number | (n=154) | (n=165) | (n=147) | (n=175) | (n=249) | (n=123) | (n=210) | (n=238) | (n=214) | (n=168) |
| | Median | 168 | 145 | 120 | 138 | 77 | 154 | 108 | 127 | 124 | 90 |
| | 75th percentile | 304 | 248 | 184 | 235 | 146 | 307 | 203 | 224 | 251 | 182 |
| | 90th percentile | 365 | 365 | 326 | 365 | 248 | 365 | 312 | 365 | 365 | 357 |
| Total interval (All patients) | Number | (n=222) | (n=221) | (n=223) | (n=209) | (n=249) | (n=148) | (n=224) | (n=238) | (n=262) | (n=205) |
| | Median | 128 | 112 | 103 | 111 | 77 | 136 | 102 | 127 | 105 | 74 |
| | 75th percentile | 239 | 201 | 159 | 211 | 146 | 266 | 194 | 224 | 230 | 153 |
| | 90th percentile | 365 | 365 | 253 | 365 | 248 | 365 | 307 | 365 | 365 | 320 |

In Sweden, no data on the primary care interval was available.

## Sensitivity and validity analyses

Changing the cut-off survey completion date from 9 months post-diagnosis to the per-protocol 6 months changed the number of included patients. However, the estimates of routes to diagnosis and time intervals were not significantly altered and the trend was the same as in the main analyses (results not shown). Sensitivity analysis based solely on patient data for those whose reporting on all time intervals was complete did not change the trend (results not shown). Comparing patient and PCP reported routes (screening and symptomatic presentation) and registry data on screening showed that agreement ranged from moderate to almost perfect (kappa 0.59–0.86).

Comparing the dates between the different data sources showed a high agreement between all data sources for all categories of dates (CCC ≥ 0.95 for date of diagnosis, CCC=0.94 for date of treatment and CCC=0.92 for date of first presentation to primary care). The analysis of the ICBP M4 definition of screening route compared with registry data showed an almost perfect agreement (kappa >0.80) in two jurisdictions and substantial (kappa >0.70) in two jurisdictions (table 3).

## DISCUSSION

Our study showed marked variation in the proportion of screen-detected cancers, lengths of diagnostic and treatment (and total) intervals between jurisdictions. Patient intervals were shortest in Denmark and longest in Wales; longer primary care intervals were present in Norway, Scotland, Manitoba and Victoria. Differences in primary care intervals do not necessarily reflect PCP delay—they may arise from PCPs undertaking more investigations prior to referral. Overall, the differences are marked and suggest the need, in some jurisdictions, for revised diagnostic pathways to reduce the time taken for patients to be diagnosed and treated.

The interval differences did not show an obvious association with earlier reported survival differences—jurisdictions with poorer survival did not consistently show longer intervals, and vice versa.[4] While this may question the validity of our findings, and/or the relationship between diagnostic intervals and survival, it is important to note these analyses were several years apart, and there may have been significant health system changes since the survival comparisons. Nevertheless, our study adds to a growing body of evidence on routes to diagnosis and time intervals; there are few similar examples involving multiple countries in the literature.[29]

Many factors underpin the differences observed between jurisdictions, such as structural differences in healthcare delivery (eg, care pathways, availability and accessibility of diagnostic and treatment facilities, etc.). Differences in routes to diagnosis were influenced by the presence or absence of screening programmes (CRC-screening had not been implemented in Denmark and Sweden at the time of study). Patient interval variation may reflect differences in symptom awareness or health-seeking behaviour. However, a study which compared awareness of cancer symptoms, attitudes towards cancer and barriers to attending a PCP did not demonstrate statistically significant differences.[30] It is possible that other factors, such as culture, rurality, economic or patient-specific barriers and facilitators, influence this important part of the pathway.[31]

There were significant differences in primary care intervals; overall these intervals were much longer than those found in our breast cancer analyses.[32] This raises important questions about diagnostic processes within primary care. There are widespread calls for PCPs to play a greater role in improving outcomes in CRC.[33] Indeed, it is now widely acknowledged that primary care has a major role in cancer control at all stages of the patient journey.[34] Nevertheless, there is some evidence based on observational associations that prolonged intervals might be associated with stronger 'gatekeeper' systems.[35]

**Table 6** Analyses of the differences in intervals (days) between Wales as the reference and the other nine jurisdictions

| | | Wales | England | Scotland | Northern Ireland | Denmark | Manitoba | Norway | Sweden | Ontario | Victoria |
|---|---|---|---|---|---|---|---|---|---|---|---|
| Patient interval (Symptomatic patients) | Number | (n=195) | (n=199) | (n=175) | (n=199) | (n=292) | (n=134) | (n=240) | (n=266) | (n=230) | (n=199) |
| | Median | 49 (ref.) | −6 (−25,13) | −14 (−29,2) | −11 (−42,20) | −25 (−38,−11) | −11 (−30,9) | −9 (−46,27) | −7 (−21,7) | −13 (−30,4) | −23 (−32,−15) |
| | 75th percentile | 92 (ref.) | 14 (−9,38) | −28 (−44,−12) | −13 (−32,6) | −28 (−47,−9) | 1 (−28,30) | −3 (−22,16) | 0 (−29,29) | 11 (−8,29) | −30 (−51,−9) |
| | 90th percentile | 249 (ref.) | 23 (−14,60) | −74 (−124,−24) | 26 (10,42) | −60 (−174,55) | −17 (−65,31) | −43 (−85,0) | −43 (−60,−27) | 3 (−109,115) | −33 (−87,21) |
| Primary care interval (Symptomatic patients) | Number | (n=157) | (n=152) | (n=127) | (n=160) | (n=207) | (n=72) | (n=124) | (n=0) | (n=77) | (n=117) |
| | Median | 3 (ref.) | −2 (−5,2) | 1 (−4,6) | −3 (−5,0) | −2 (−5,1) | 0 (−8,8) | 7 (3,12) | n/a | −2 (−5,1) | 6 (0,13) |
| | 75th percentile | 20 (ref.) | −1 (−14,12) | 9 (−6,25) | −7 (−20,6) | −10 (−25,4) | 9 (−3,21) | 21 (3,39) | n/a | 3 (−8,15) | 13 (−6,31) |
| | 90th percentile | 78 (ref.) | −14 (−31,3) | 42 (11,73) | −23 (−64,17) | −19 (−91,53) | 124 (63,186) | 36 (0,71) | n/a | 3 (−43,49) | 36 (−16,88) |
| Diagnostic interval (Symptomatic patients) | Number | (n=194) | (n=196) | (n=174) | (n=190) | (n=290) | (n=133) | (n=229) | (n=249) | (n=218) | (n=197) |
| | Median | 60 (ref.) | −11 (−18,−4) | −20 (−27,−13) | 5 (−6,16) | −29 (−35,−24) | 14 (−1,29) | −20 (−28,−12) | −19 (−35,−3) | −1 (−11,8) | −28 (−35,−21) |
| | 75th percentile | 155 (ref.) | −56 (−119,7) | −59 (−113,−5) | −44 (−81,−7) | −83 (−110,−55) | −6 (−60,47) | −67 (−93,−42) | −65 (−117,−13) | −3 (−52,46) | −81 (−131,−32) |
| | 90th percentile | 284 (ref.) | −78 (−116,−40) | −114 (−132,−96) | −54 (−89,−19) | −130 (−161,−99) | 33 (9,56) | −80 (−114,−46) | −86 (−116,−56) | 15 (−2,32) | −78 (−103,−53) |
| Diagnostic interval (Screen-detected patients) | Number | (n=69) | (n=56) | (n=76) | (n=35) | (n=0) | (n=25) | (n=14) | (n=0) | (n=50) | (n=38) |
| | Median | 35 (ref.) | −8 (−13,−4) | 3 (−2,8) | −32 (−36,−28) | n/a | 31 (19,43) | −15 (−24,−5) | n/a | −25 (−34,−15) | 1 (−8,10) |
| | 75th percentile | 65 (ref.) | −18 (−28,−8) | −12 (−29,4) | −63 (−70,−55) | n/a | 36 (6,66) | −24 (−42,−6) | n/a | −17 (−26,−8) | 6 (−20,33) |
| | 90th percentile | 99 (ref.) | −17 (−28,−6) | 12 (−16,41) | −98 (−101,−96) | n/a | 90 (32,147) | −35 (−43,−27) | n/a | 40 (30,51) | 32 (25,40) |
| Diagnostic interval (All patients) | Number | (n=263) | (n=252) | (n=250) | (n=225) | (n=290) | (n=158) | (n=243) | (n=249) | (n=268) | (n=235) |
| | Median | 52 (ref.) | −9 (−15,−4) | −13 (−19,−7) | −2 (−12,7) | −22 (−28,−17) | 21 (13,29) | −14 (−24,−3) | −13 (−21,−4) | −4 (−11,2) | −21 (−26,−15) |
| | 75th percentile | 120 (ref.) | −34 (−49,−18) | −38 (−53,−23) | −10 (−31,10) | −44 (−56,−32) | 22 (4,41) | −29 (−50,−9) | −27 (−38,−17) | 18 (2,34) | −46 (−57,−34) |
| | 90th percentile | 242 (ref.) | −73 (−101,−45) | −91 (−118,−64) | −41 (−57,−24) | −106 (−127,−85) | 50 (7,94) | −53 (−88,−18) | −59 (−88,−30) | 44 (27,61) | −54 (−82,−26) |
| Treatment interval (Symptomatic patients) | Number | (n=197) | (n=206) | (n=185) | (n=208) | (n=306) | (n=161) | (n=258) | (n=281) | (n=248) | (n=209) |
| | Median | 39 (ref.) | −6 (−11,0) | −5 (−11,0) | −13 (−19,−8) | −24 (−27,−21) | −5 (−14,5) | −20 (−23,−16) | −3 (−8,2) | −6 (−10,−1) | −23 (−28,−19) |
| | 75th percentile | 59 (ref.) | −13 (−18,−8) | −4 (−11,4) | −19 (−24,−13) | −41 (−48,−34) | 0 (−9,10) | −30 (−37,−24) | −8 (−16,−1) | −6 (−12,0) | −31 (−38,−25) |
| | 90th percentile | 83 (ref.) | −27 (−38,−16) | −5 (−20,11) | −29 (−42,−17) | −61 (−72,−50) | 4 (−13,20) | −43 (−59,−27) | −24 (−35,−14) | −8 (−21,4) | −42 (−52,−31) |
| Treatment interval (Screen-detected patients) | Number | (n=72) | (n=58) | (n=78) | (n=34) | (n=0) | (n=79) | (n=17) | (n=0) | (n=52) | (n=60) |
| | Median | 44 (ref.) | −4 (−14,5) | 3 (−6,12) | −5 (−15,5) | n/a | −8 (−17,2) | −28 (−41,−16) | n/a | −8 (−14,−2) | −26 (−36,−16) |
| | 75th percentile | 68 (ref.) | −22 (−26,−19) | 0 (−5,5) | −22 (−25,−18) | n/a | −13 (−22,−3) | −45 (−49,−42) | n/a | −23 (−26,−20) | −37 (−45,−29) |
| | 90th percentile | 80 (ref.) | −21 (−32,−9) | 6 (−3,14) | −23 (−31,−15) | n/a | −6 (−15,3) | −37 (−45,−30) | n/a | 11 (−1,22) | −42 (−51,−33) |
| Treatment interval (All patients) | Number | (n=271) | (n=268) | (n=266) | (n=246) | (n=312) | (n=240) | (n=279) | (n=284) | (n=310) | (n=276) |
| | Median | 41 (ref.) | −6 (−10,−2) | −2 (−9,5) | −13 (−18,−9) | −26 (−30,−21) | −4 (−10,1) | −22 (−27,−16) | −5 (−10,0) | −6 (−11,−1) | −24 (−29,−20) |
| | 75th percentile | 63 (ref.) | −15 (−21,−8) | 2 (−10,14) | −19 (−27,−11) | −42 (−50,−35) | 0 (−7,7) | −32 (−39,−24) | −9 (−16,−1) | −8 (−17,2) | −32 (−46,−17) |
| | 90th percentile | 80 (ref.) | −24 (−31,−17) | 3 (−12,18) | −25 (−34,−17) | −59 (−66,−51) | 4 (−4,12) | −41 (−54,−29) | −22 (−30,−15) | −4 (−19,11) | −40 (−50,−30) |
| Total interval (Symptomatic patients) | Number | (n=154) | (n=165) | (n=147) | (n=175) | (n=249) | (n=123) | (n=210) | (n=238) | (n=214) | (n=168) |
| | Median | 168 (ref.) | −30 (−55,−6) | −52 (−99,−5) | −36 (−93,21) | −92 (−106,−78) | −25 (−86,35) | −63 (−74,−52) | −43 (−63,−24) | −44 (−98,9) | −85 (−97,−73) |
| | 75th percentile | 304 (ref.) | −35 (−78,8) | −101 (−130,−71) | −34 (−91,22) | −137 (−177,−97) | −10 (−24,4) | −84 (−129,−38) | −62 (−92,−32) | −44 (−91,2) | −117 (−137,−97) |
| | 90th percentile | 365 (ref.) | 0 (−1,0) | −39 (−47,−32) | 0 (−1,0) | −125 (−134,−116) | 0 (−1,0) | −49 (−58,−41) | 0 (−1,0) | 0 (−1,0) | −7 (−16,1) |

Continued

| Table 6 Continued | | Wales | England | Scotland | Northern Ireland | Denmark | Manitoba | Norway | Sweden | Ontario | Victoria |
|---|---|---|---|---|---|---|---|---|---|---|---|
| Total interval (All patients) | Number | (n=222) | (n=221) | (n=223) | (n=209) | (n=249) | (n=148) | (n=224) | (n=238) | (n=262) | (n=205) |
| | Median | 128 (ref.) | −13 (−35,9) | −22 (−36,−8) | −14 (−38,10) | −46 (−62,−30) | 10 (−13,32) | −19 (−34,−5) | −1 (−16,15) | −18 (−34,−2) | −44 (−70,−18) |
| | 75th percentile | 239 (ref.) | −32 (−63,−1) | −81 (−126,−35) | −26 (−64,12) | −89 (−137,−41) | 10 (−33,52) | −43 (−70,−17) | −18 (−59,23) | −16 (−80,49) | −88 (−119,−57) |
| | 90th percentile | 365 (ref.) | −1 (−2,0) | −108 (−129,−88) | −1 (−3,1) | −118 (−127,−109) | 0 (−2,2) | −56 (−80,−32) | 0 (−1,0) | 0 (−2,1) | −46 (−54,−37) |

The differences for the 50th (median), 75th and 90th percentiles are calculated as marginal effects after quantile regression by setting the continuous covariate age to its mean value and categorical covariates (gender and comorbidity) to their modes. The actual number of days are included for Wales in table 5.
n/a, not applicable; ref. reference.

Further, independent of the nature of symptoms, investigation in primary care has been noted to be associated with later referral for specialist assessment for CRC and other cancers—so a long primary care interval may mean that PCPs are doing more before they refer.[36] Hence, PCP access to and use of investigations and differing national cancer referral guidance may influence primary care intervals. There were differences in diagnostic intervals, suggesting that once patients have been referred to secondary care there is considerable variation in their experiences; differences in treatment intervals were less marked. These variations suggest there is room for improvement in reducing the total interval and its various subintervals, and that guidance on optimal pathways should be better implemented. Each participating jurisdiction will likely draw unique conclusions about the most appropriate response to our findings.

### Strengths and weaknesses

A major strength of this study is its use of standardised survey methods in a broad range of jurisdictions to systematically examine the various components of these intervals and to describe and compare, between countries, patient journeys to a cancer diagnosis and treatment. To ensure comparability across jurisdictions, our surveys drew on existing instruments and went through an extensive process of cognitive testing, piloting, and translation and adaptation.[18]

Data quality was enriched by information from national cancer registries and our algorithms showed very good agreement for jurisdictions where validation was possible. Using validated identification of patients with CRC minimised the risk of missing cancer cases during inclusion and of selection bias. Further, the use of registries made it possible to exclude patients with previous cancer in the same site, providing a homogeneous group of newly diagnosed patients with CRC in need of diagnostic work-up.

It was evident that there were subtle differences in the understanding of 'screening' between jurisdictions. Patients do not always distinguish between tests for screening and those for symptom-based diagnosis. Including data from registries and triangulating patient and PCP data enhanced the validity of 'screen-detected versus non-screen-detected' information, but the underlying factors varied between jurisdictions—for example, in Australia PCPs often provide screening FOBTs during consultations whereas this is rare in the UK and Scandinavia. To counter these inconsistencies, we applied our validated data rules which showed a high agreement with screening registries.

There are inevitable differences in questionnaire interpretation, characteristics of non-responders and availability of supplementary data for validation purposes. There are always considerations with questionnaire interpretation but the methodology and analysis of data sought to minimise or account for this as much as possible. Further, we used triangulation and comprehensive data rules to ensure validity, consistency and

preserve statistical precision.[18] We included registry data where possible (screening, stage, date of diagnosis) and developed reliable rules for imputation based on these registry data. To minimise misclassification from data entry and handling, data entry was internally audited by local teams and data interpretation was reduced to an absolute minimum and only performed centrally. Recall bias was minimised by the triangulation of different data sources and by ensuring that the patients received the questionnaire with a limited time window after the cancer diagnosis.

The overall response of 31% for the patient survey varied between jurisdictions. There were likely differences in the selection bias in individual jurisdictions; our patient sampling strategy will have led to some differences in the composition of our samples, as some patients were included directly from registries, some via PCPs and, in Northern Ireland, research nurses checked lists of potentially eligible patients to confirm eligibility. We have no mechanism to examine the direction of such possible selection bias. However, comparison of participating patients on a number of variables (including comorbidity, self-assessed health, smoking, stage, presenting symptom) did not show obvious differences with potential to bias our results. We also compared age, sex and stage of cancer among participants versus eligible patients and found no significant differences. There were different classification systems for ethnicity and education across jurisdictions which would lead to biased estimates if included in the regression model, even if mapped or harmonised—hence, they were excluded. There were few respondents from minority ethnic groups, limiting the generalisability of our findings; further work should target these groups as they are likely to have unique characteristics in their routes to diagnosis.

Confounding from aspects related to the diagnostic route for CRC was diminished by adjusting for age, gender and comorbidity. It is possible that there is some residual confounding which can bias the results in different directions. The statistical precision of the study was high as we were able to show clinically significant differences of 1 week in time intervals.

### Comparison with other studies

Other studies have examined symptoms and routes to diagnosis for patients with CRC—although rarely in more than a single setting. A UK study on patients diagnosed in 2001–2002 and 2007–2008 reported median diagnostic intervals of 100 and 80 days, respectively.[37] A Danish study showed median diagnostic intervals for CRC of between 31 and 55 days, depending on the timing of measurement in relation to implementation of pathway guidance.[38] The difference between the present study and the former may stem from methodology issues, especially data sources (ie, databases vs surveys).

A study in Spain showed a symptom to diagnosis interval for CRC of 128 days and symptom to treatment interval of 155 days—these authors found that nature of symptoms, perceived seriousness of symptoms by patients, and place of first presentation influenced diagnostic and treatment intervals.[39] Sampling strategies and survey differences will have influenced the results, making it difficult to compare these studies; nevertheless, they confirm that our results are broadly consistent with previous, single-jurisdiction studies.

The pattern of symptoms in ICBP participants was similar to other studies.[40] CRC is known to be a cancer that clinically presents with either 'alarm' symptoms or more vague symptoms; there is evidence that doctors and patients respond less promptly to some symptoms of CRC than others—and that this can be influenced by the presence of co-morbid conditions.[41] However, the minor differences in symptom patterns seen in table 4 are insufficient to explain the between-jurisdiction variation we have demonstrated in routes to diagnosis and diagnostic, treatment and total intervals.

### Explaining observed differences between jurisdictions

The variation we see between jurisdictions mostly derives from differences in the extent to which healthcare systems support expedited CRC diagnosis and treatment; indeed, some health system characteristics, such as access and patient mobility between healthcare providers, may influence cancer outcomes—although these factors require further exploration.[42] In Denmark there have been a number of reforms specifically designed to reduce diagnostic intervals.[43] This study indicates a potential to optimise diagnostic routes for CRC in some jurisdictions. This should ideally be in conjunction with screening efforts which is gaining traction across many Western countries in response to policy and guideline initiatives.[44]

### CONCLUSION

This study demonstrates considerable absolute and relative differences between jurisdictions in time intervals from first symptom until treatment for CRC. These differences do not demonstrate an obvious relationship with existing ICBP survival differences between the jurisdictions. ICBP phase 2 will report survival estimates to 2014, at which point it may be possible to explore the relationship between interval lengths and survival estimates in the population. Further work is ongoing to explore the outcome of patients included in this ICBP M4 study alongside the interval lengths observed.

The median total interval, which varied between 74 and 136 days, indicates that there is unrealised potential to optimise pathways for CRC. The main differences were found for structural parts of the pathway (eg, those not relating to patient behaviours/actions). Further, there is a 'tail' of patients waiting many months longer to start treatment for their cancer which may affect their outcomes. While our study highlights important international differences in routes to diagnosis, further research is needed to understand these differences and elucidate the contribution of patient pathway guidance and

implementation, and health system structures. Nevertheless, the data provide important prompts for jurisdictions and suggest considerable room for improvement in some areas; they will also serve as a benchmark for measuring the effectiveness of future interventions.

**Author affiliations**
[1]MRC Clinical Trials Unit and Instittue of Clinical Trials and Methodology, University College London, London, UK
[2]Research Unit for General Practice, Aarhus University, Aarhus, Denmark
[3]Prevention and Cancer Control, Cancer Care Ontario, Toronto, Ontario, Canada
[4]European Palliative Care Research Centre (PRC), Department of Oncology, Oslo University Hospital and Institute of Clinical Medicine, University of Oslo, Oslo, Norway
[5]Centre for Behavioral Research in Cancer, Cancer Council Victoria, Melbourne, Victoria, Australia
[6]Department of General Practice, University of Melbourne, Melbourne, Australia
[7]Centre for Population Health Sciences, University of Edinburgh, Edinburgh, UK
[8]Scottish Cancer Registry, Information Services Division, NHS National Services Scotland, Edinburgh, UK
[9]Northern Ireland Cancer Registry, Queen's University Belfast, Belfast, UK
[10]Health Services Research Program, Ontario Institute for Cancer Research, Toronto, Ontario, Canada
[11]Department of Epidemiology and Cancer Registry, Cancer Care Manitoba, Winnipeg, Manitoba, Canada
[12]Regional Cancer Center Uppsala and Department of Medical Epidemiology and Biostatics, Karolinska Institutet, Stockholm, Sweden
[13]North Wales Centre for Primary Care Research, Bangor Institute for Health and Medical Research, Bangor University, Bangor, UK
[14]Department of Oncology, Lund University Hospital, Lund, Sweden
[15]Population Oncology, Cancer Care Manitoba, Winnipeg, Canada
[16]Academic Unit of Primary Care, University of Leeds, Leeds, UK
[17]School of Psychology, Deakin University School of Psychology, Burwood, Victoria, Australia
[18]Policy and Information, Cancer Research UK, London, UK

**Acknowledgements** We thank our colleagues and participating patients, primary care physicians and cancer treatment specialists in all ICBP jurisdictions. Catherine Foot, Martine Bomb, Brad Groves, Irene Reguilon and Samantha Harrison of Cancer Research UK for managing the programme. John Butler, Royal Marsden NHS Foundation Trust and ICBP Clinical Lead, for his advice. The ICBP M4 academic reference group (online supplementary file 2) for providing independent peer review and advice for the study protocol and analysis plan development. Throughout the study, invaluable guidance was provide by the ICBP Programme Board. Membership of this board: Aileen Keel (Scottish Government, Edinburgh, Scotland); Anna Boltong (Cancer Council Victoria, Carlton, Australia); Anna Gavin (Northern Ireland Cancer Registry, Queens University, Belfast, UK); David Currow (Cancer Institute New South Wales, Sydney, Australia); Gareth Davies (Wales Cancer Network, Cardiff, UK); Gunilla Gunnarsson (Swedish Association of Local Authorities and Regions, Stockholm, Sweden); Heather Bryant (Canadian Partnership Against Cancer, Toronto, Canada); Jane Hanson (Welsh Cancer National Specialist Advisory Group, Cardiff, UK); Kathryn Whitfield (Department of Health, Victoria Australia); Linda Rabeneck (Cancer Care Ontario, Toronto, Canada); Michael A Richards (Care Quality Commission, London, UK); Michael Sherar (Cancer Care Ontario, Toronto, Canada); Nicola Quin (Cancer Council Victoria, Carlton, Australia); Nicole Mittmann (Cancer Care Ontario, Toronto, Canada); Robert Thomas (Department of Health and Human Services, Victoria, Melbourne, Australia); Sara Hiom (Cancer Research UK); Sean Duffy (NHS England, London, UK); Chris Harrison (NHS England, London, UK); Søren Brostrøm (Danish Health and Medicines Authority, Copenhagen, Denmark); Stein Kaasa (University Hospital of Trondheim, Trondheim, Norway).

**Contributors** DW, PV, UM, AZF and HJ planned the study design, data collection, carried out the analyses and wrote the draft manuscript. DW, PV, UM, HJ, AB, AKK, RB, DHB, VC, ATG, EG, EH, ML, R-JL, YL, MM, DT, RDN, VW, IR and SH were responsible for local data collection (alongside the ICBPM4WG), management and interpretation, and have participated in writing and approving the final manuscript version.

**Funding** This work was supported by: CancerCare Manitoba; Cancer Care Ontario; Canadian Partnership Against Cancer (CPAC); Cancer Council Victoria; Cancer Research Wales; Cancer Research UK; Danish Cancer Society; Danish Health and Medicines Authority; European Palliative Care Research Centre (PRC), Norwegian University of Science and Technology (NTNU); Northern Ireland Guidelines Audit and Implementation Network (GAIN); Macmillan Cancer Support; National Cancer Action Team; NHS England; Northern Ireland Cancer Registry, funded by the Public Health Agency NI; Norwegian Directorate of Health; Research Centre for Cancer Diagnosis in Primary Care (CaP), Aarhus University, Denmark; Scottish Government; Swedish Association of Local Authorities and Regions; University College London and NIHR Biomedical Research Centre at University College London NHS Foundation Trust; University of Edinburgh; Victorian Department of Health and Human Services; Welsh Government.

**Disclaimer** The funding bodies had no influence on the design of the study and collection, analysis and interpretation of data, in writing the manuscript or whether to publish the results.

**Competing interests** None declared.

**Patient consent** Not required.

**Ethics approval** For each local data collection, there were specific procedures and approvals which included anonymised data transfer to University College London and Aarhus University (online supplementary file 5). Approvals were received from the following institutions: Cancer Council Victoria Human Research Ethics Committee [HREC 1125]; Health Research Ethics Board, University of Manitoba [HS15227 (H2012:105)]; Research Resource Ethics Committee, CancerCare Manitoba [RRIC#28-2012]; University of Toronto Research Ethics Board [27881]; The Danish Data Protection Agency [2013-41-2030]; Swedish Ethics Review Board, Uppsala [2013/306]; Norway Regional committees for medical and health research ethics [2013/136/REK nord]; England, Wales and Scotland, NRES Committee East Midlands – Derby 2, local R&D for each health board [11/EM/0420]; Northern Ireland ORECNI Ethical approval, local governance for each health Trust [11/EM/0420].

**Provenance and peer review** Not commissioned; externally peer reviewed.

**Data sharing statement** The data that support the findings of this study are available from the named authors from each ICBP jurisdiction, but restrictions apply to the availability of these data and so are not publicly available. Data are however available from the authors on reasonable request and with permission of the ICBP Programme Board. Please contact the ICBP Programme Management team, based at Cancer Research UK, with any queries (icbp@cancer.org.uk).

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
