## [Reviewer comments · BMJ Open]

ARTICLE DETAILS

TITLE (PROVISIONAL)	Diagnostic routes and time intervals for colorectal cancer patients in ten international jurisdictions; findings from a cross-sectional study from the International Cancer Benchmarking Partnership (ICBP)
AUTHORS	Weller, David; Menon, Usha; Zalounina Falborg, Alina; Jensen, Henry; Barisic, Andriana; Knudsen, Anne Kari; Bergin, Rebecca; Brewster, David; Cairnduff, Victoria; Gavin, Anna; Grunfeld, Eva; Harland, Elizabeth; Lambe, Mats; Law, Rebecca-Jane; Lin, Yulan; Malmberg, Martin; Turner, Donna; Neal, RD; White, Victoria; Harrison, Samantha; Reguilon, Irene; Working Group, ICBP; Vedsted, Peter

VERSION 1 – REVIEW

REVIEWER	Allan Spigelman UNSW St Vincents Clinical School, Sydney, Australia
REVIEW RETURNED	07-May-2018

GENERAL COMMENTS	This paper it is well written (apart from "jurisdictions" in the abstract)
--

REVIEWER	Prof Danielle Mazza Monash University, Australia
REVIEW RETURNED	20-May-2018

GENERAL COMMENTS	Thanks for the opportunity to review this interesting paper providing an international comparison of diagnostic routes and time intervals for colorectal cancer in 10 international jurisdictions. The paper is generally well written but I think could be improved by addressing the following comments below: Background:  • CRC is the second most common cause of cancer death in Europe after what? • Please introduce what the ICBP is before discussing module 4 • Line 28 page 5 better to say it may help shape policy rather than this will help shape policy • Page 6:  o Line 4, refer to jurisdiction then country where relevant o Line 8 – to measure the patient..... o Line 22 – were the surveys the same? Who developed these and what were they based on – please expand o Line48 – registries via the research team who? o Who sent the pps and csts the letter, did patients give consent for them to be contacted? • Page 7  o Line 25 – how were they validated? • Page 10
---

	 o How were patients involved in the piloting? Did this occur in each jurisdiction? Please expand  • Page 12 o Why were most of the screen detected cancers in northern Ireland excluded in the recruitment process? o Why are data not shown (line 35)?  • Page 14 o Line 35 and 36 should go in methods section Discussion o I think it is important to expand on the points raised about the lack of consistency between survival differences and longer intervals as it does question the validity and utility of the findings o When explaining observed differences between jurisdictions it would be good to have a table which provides a contextual description of relevant issues in each jurisdiction
--	--

REVIEWER	Dr Andrew Veitch Royal Wolverhampton NHS Trust, England
REVIEW RETURNED	24-May-2018

GENERAL COMMENTS	This is an ambitious benchmarking study across several jurisdictions examining time intervals to diagnosis of colorectal cancer. A poor response rate of 31%, and under-representation of certain ethnic minorities weakens the study. It is hypothesised in the introduction that the poor outcomes in Denmark and UK may be related to delays in diagnosis, but this is not borne out by the results. Neither does there appear to be a relation between delays in presentation with symptoms and outcomes including cancer stage at diagnosis between the study groups. With the achieved response and sample size, was the study statistically powered to demonstrate differences for each of the parameters studied? Where there differences in delays in between the date of diagnosis and patients responding to a questionnaire? This could result in recall bias if the delays are long. It is also possible that some patients with advanced disease could have died before having the opportunity to participate. The date relates to a time period up to 2007. Are there more recent data from any of the jurisdictions with which to compare this with and demonstrate any change or trends?
--

REVIEWER	Magdalena Esteva Unit of Research, Majorca Primary Care District, Institute of Health Research of the Balearic Islands
REVIEW RETURNED	29-May-2018

GENERAL COMMENTS	Congratulations to the authors for this interesting paper on the comparison of international differences of routes of diagnosis for CRC patients and time intervals from symptom onset until the start of treatment. This is a well conducted study with interesting results. There are some points in the discussion section that deserve some major comments: First of all, in the first paragraph of the discussion, authors declare that interval differences did not show an obvious correlation with earlier reported survival differences. But as far as I am concerned no correlation analysis has been done of different survival figures in the jurisdictions with time intervals. Please clarify the statement.
---

	In spite of the low response rate by patients and PCPs, it seems that the distribution of age, sex and stage were similar between eligible patients and participating patients and authors have clarified the possible and in consequence not very relevant selection bias... I wonder if a matter of relevance are the differences in treatment between the different jurisdictions. Have patients of the different jurisdictions the same access to the best evidenced treatment? or they are submitted to different constraints depending of health system delivery or health assurance. It will be interesting if the authors could add any published information on the subject. Good job
--	--

VERSION 1 – AUTHOR RESPONSE

Thanks to Reviewer 1 for the suggestion to have a table which provides contextual descriptions of the relevant issues in ICBP jurisdictions. Previous work carried out in ICBP Module 3 - Brown et al, reference 42 in the manuscript - included a mapping of health system differences and included a table of the key characteristics of primary care. As such, we have decided not to include a further table but have noted this reference more clearly in the manuscript.

Thanks to Reviewer 2 for the suggestion that we clarify that colorectal cancer is the second most common cancer after female breast cancer, and that we provide more information on the aims of the International Cancer Benchmarking Partnership before introducing the aims of Module 4. We now explain more about ICBP and lead the reader to further published work. We have also moved page 14 lines 35 and 36 to the methods section as suggested.

Similarly, we have made sure to clarify that there is a publication which describes the development and validation of the surveys used in this work, as well as patient involvement in this process, and have referenced appropriately. We have also clarified that patients needed to consent to surveys being sent to their primary care and specialist physicians. We have also clarified that in Northern Ireland, the study team actively monitored the route (screened vs symptomatic) of patients returning surveys – and prioritised the responses from symptomatic patients for inclusion in the study as there was a 200-symptomatic patient target per jurisdiction.

We thank Reviewers 2, 3 and 4 for raising questions about survival and interval differences which is also a very interesting outcome. Further analyses are ongoing to explore associations between the time interval variation observed in this study and outcomes for patients involved. Other outcomes like stage, patient experience and costs are important. Thus, we see the findings presented, and the substantial variation in time intervals experienced by patients internationally, as important for health policymakers and clinicians. Our findings show that health system organisation could in some jurisdictions be optimised, and highlights jurisdictions which may have a more beneficial and expedited process for the diagnosis of colorectal cancer. We hope that this leads to further work within and between jurisdictions to further understand barriers and enablers to the timely diagnosis of patients through both primary and secondary care, as well as a consideration of public awareness of potential colorectal cancer symptoms.

As the survival estimates from the ICBP underpin the explanatory work that includes ICBP Module 4, we have made reference to the survival estimates from ICBP Module 1. However, this work covers survival estimates up to 2007, which makes the comparison between our reported intervals (2013-15) and the reported survival in the Coleman et al paper difficult. Many health system factors have changed since then, and more emphasis on more timely diagnosis is one example. We have added a line to explain that this work may be something to explore when ICBP phase 2 reports survival estimates. We have also clarified that work to explore outcomes and interval lengths in this cohort is planned, as we have collected status at one year.

Thanks to Reviewer 3 for questions about delays in responding to the questionnaire. We have added a row with interval for diagnosis to consent (Table 2) and a line in the Results section to clarify this. We agree that recall bias may be of concern in studies as this. Therefore, we undertook a sensitivity analysis only using questionnaires returned within six months of diagnosis. This did not change the results and is reported in the paragraph "Sensitivity and validity analyses" in the results section. We thank Reviewer 4 for agreeing that the distribution of the patients identified and participating are similar, and therefore not likely to have led to selection bias, although response rates were low and there was a low proportion of ethnic minorities, also raised by Reviewer 2. We checked against the population and the respondents did differ as expected by age, vital status and tumour stage. Respondents were younger, had lower tumour stage and were more likely to be alive one-year after diagnosis – all factors had p-values below 0.001 (and sex p=0.076). We thank Reviewer 4 for highlighting treatment differences. One thing is that there was less variation observed in the treatment interval internationally, compared to the diagnostic interval. The jurisdictions included in the ICBP were selected as they had similar level of cancer treatment. Although there will be some differences in treatment guidelines, access and adherence, this would not create such differences in time intervals shown. Lastly, thanks to all reviewers and the editorial team for highlighting typos and missing words, we are grateful for these being pointed out and have corrected and reviewed the manuscript.

VERSION 2 – REVIEW

REVIEWER	Dr Andrew Veitch Royal Wolverhampton NHS Trust, England
REVIEW RETURNED	07-Aug-2018
GENERAL COMMENTS	Thank you for addressing the reviewers' comments
REVIEWER	Magdalena Esteva Majorca Primary Care Department, Spain
REVIEW RETURNED	21-Aug-2018
GENERAL COMMENTS	I think authors have answered to the comments of reviewers adequately, and I think the manuscript is of high value for the Journal.